# Systematic In Silico Assessment of Antimicrobial Resistance Dissemination across the Global Plasmidome

**DOI:** 10.3390/antibiotics12020281

**Published:** 2023-02-01

**Authors:** Miquel Sánchez-Osuna, Jordi Barbé, Ivan Erill

**Affiliations:** 1Departament de Genètica i de Microbiologia, Universitat Autònoma de Barcelona, 08193 Bellaterra, Spain; 2Department of Biological Sciences, University of Maryland Baltimore County, Baltimore, MD 21250, USA

**Keywords:** antimicrobial resistance, dissemination, evolution, plasmid, mobile genetic element, GC content, horizontal gene transfer

## Abstract

The emergence of pathogenic strains resistant to multiple antimicrobials is a pressing problem in modern healthcare. Antimicrobial resistance is mediated primarily by dissemination of resistance determinants via horizontal gene transfer. The dissemination of some resistance genes has been well documented, but few studies have analyzed the patterns underpinning the dissemination of antimicrobial resistance genes. Analyzing the %GC content of plasmid-borne antimicrobial resistance genes relative to their host genome %GC content provides a means to efficiently detect and quantify dissemination of antimicrobial resistance genes. In this work we automate %GC content analysis to perform a comprehensive analysis of known antimicrobial resistance genes in publicly available plasmid sequences. We find that the degree to which antimicrobial resistance genes are disseminated depends primarily on the resistance mechanism. Our analysis identifies conjugative plasmids as primary dissemination vectors and indicates that most broadly disseminated genes have spread from single genomic backgrounds. We show that resistance dissemination profiles vary greatly among antimicrobials, oftentimes reflecting stewardship measures. Our findings establish %GC content analysis as a powerful, intuitive and scalable method to monitor the dissemination of resistance determinants using publicly available sequence data.

## 1. Introduction

The discovery of antibacterial compounds is considered one of the great advances in modern medicine, since it has paved the way for the effective treatment of infections caused by pathogenic bacteria. However, these agents have been losing their efficacy due to the emergence and dissemination of resistance among bacterial pathogens [1]. In this sense, the number of infections caused by multi-resistant bacteria is increasing globally, making some of them untreatable. This has prompted the World Health Organization (WHO) to classify antibacterial resistance as one of the three most important public health threats of the 21st century [2].

The development of resistance to antibacterial compounds is associated with their misuse and abuse [3]. The high genetic plasticity of bacteria enables them to rapidly adapt to a wide range of environmental threats, including the one posed by antibacterial agents. In addition to the clinical environment, animal production, agriculture, aquaculture and effluents from pharmaceutical industries and municipal wastewater systems have all been postulated as hotspots for the emergence of antibacterial resistance [4]. These environments are characterized by high bacterial loads, where constant exposure to sub-inhibitory concentrations of different antimicrobial agents can promote the selection, mobilization and dissemination of genes conferring resistance to them [5]. In these environments, bacteria can leverage two alternative processes to mitigate the effects of antimicrobial agents. On the one hand, spontaneous mutations conferring resistance may occur and be selected for. These mutations may target genes directly associated with the mechanism of action of the drug, impact genes indirectly associated, such as mutations on regulatory processes or interacting partners, or confer cryptic resistance by modifying different aspects of bacterial physiology [6]. On the other hand, bacteria may acquire foreign genetic material encoding resistance to antibacterial compounds through horizontal gene transfer (HGT) [7]. Consequently, the extensive use of antimicrobial agents has led to the co-selection of resistance genes, their mobilization, dissemination and fixation, and to the increased presence of multiple resistance genes in bacterial populations in many ecosystems [8].

It is now widely accepted that the rapid proliferation of multi-resistant pathogens has not been driven by the independent evolution of mutations conferring resistance to different antimicrobials, but through the spreading of mobile genetic elements (MGE) carrying resistance genes via HGT [9,10]. Bacteria classically take up foreign DNA through transformation (direct DNA acquisition from the environment), transduction (acquisition of DNA mediated by bacteriophages), conjugation (acquisition of DNA mediated by cell-cell contact) and vesiduction (cell fusion with membrane vesicles) [11,12]. Among these mechanisms, conjugative plasmids are presumed to be the main drivers of the dissemination of antimicrobial resistance determinants, and they appear to have played an outsized role in the current global antimicrobial resistance crisis [13]. Plasmid-borne antimicrobial resistance factors can modify or protect the antimicrobial target, inactivate the antimicrobial molecule, decrease the intracellular availability of the antibacterial compound or act as alternative enzymes [9].

Several studies have shown that some plasmid-borne genes conferring resistance to antibiotics in clinical isolates have originated by successive horizontal gene transfer of homologous genes available since ancient times [8]. Recent work indicates that this is also true for genes conferring resistance to manmade chemotherapeutic agents, with resistance predating the invention and use of these antimicrobial compounds [14,15]. Therefore, the mobilization of resistance genes from different source microorganisms and their subsequent dissemination by mobile genetic elements has given rise to the multiple variants of antimicrobial resistance determinants present today in pathogenic bacteria [10].

Many studies have elucidated the factors and mechanisms that govern the dispersal of MGEs [16,17,18], and the dissemination of specific resistance genes has been thoroughly documented [8]. However, relatively few studies have addressed the principles governing the mobilization and subsequent dissemination of antimicrobial resistance genes (ARGs) [19,20]. We have previously shown that analyzing the %GC distribution of resistance genes in plasmids can highlight independent mobilization events [14,15]. Bacterial %GC content is predominantly driven by mutational biases in the replication machinery, and therefore stable among closely related species [21]. The %GC content and codon usage pattern of a horizontally transferred gene will gradually change to match its host genome. This amelioration process is driven by the mutational processes of the host. Over time, introgressed genes accumulate mutations following the host mutational biases, and may also undergo selection to match the host codon usage bias [22]. Phylogenetic evidence and mathematical modeling show that the amelioration process takes millions of years, with estimated rates of divergence in %GC content for coding sequences of 0.045% and 0.91% per million years at nonsynonymous and synonymous codons [15,22]. Plotting host chromosome versus antimicrobial resistance gene %GC content can thus generate two markedly different scenarios, as illustrated in Figure 1. If a resistance gene has been mobilized from the chromosome in different lineages, but has been transferred only among closely related species, the ARG %GC content is expected to align with the host chromosome %GC content, resulting in a diagonal plot (Figure 1A). In contrast, if a resistance gene has been rapidly and widely disseminated following one or more original mobilization events, the ARG %GC content will be essentially invariant (since amelioration takes millions of years) and hence independent from the host %GC content, presenting as one or more horizontal dissemination bands in the %GC plot (Figure 1B), as reported previously [14].

In this work, we leverage this %GC content-based analysis to systematically assess the dissemination of known resistance determinants across the plasmidome, which represents a major concern for public health [2]. Our results establish the analysis of %GC distribution as a direct and powerful method to monitor the dissemination of resistance determinants. In accordance with recent results, we find that the prevalence of ARG dissemination is determined primarily by the mechanism of action of the resistance gene, that resistance to a significant number of antimicrobial compounds has been disseminated multiple times from independent sources, and that conjugative plasmids play an outsized role in ARG dissemination. We report that different antimicrobial classes present remarkably diverse ARG dissemination profiles, and that plasmid-based ARG dissemination is not detected for many restricted-use antimicrobials.

## 2. Results and Discussion

### 2.1. Independent Mobilization and Dissemination of Antimicrobial Resistance Genes

To explore ARG dissemination across the global plasmidome, we used a protein sequence panel of representative ARG from the Comprehensive Antibiotic Resistance Database (CARD) to detect putative ARG homologs in plasmid sequences from the PLSDB database. We then compiled the nucleotide sequences of the genes encoding these proteins and computed their %GC content, as well as the chromosomal %GC content of the bacterial strain, species or genus harboring the plasmid (Appendix A). Previous studies have used Antibiotic Resistance Ontology (ARO) identifiers for individual ARG to analyze their dissemination [20], but this level of granularity arbitrarily separates protein homologs that should be considered jointly for %GC analysis. Here, we mapped ARG homologs to Clusters of Orthologous Groups (COG) identifiers and we grouped ARG homologs based on their CARD-assigned antimicrobial resistance (AMR) Gene Family and the COG they mapped to (ARO-COG groups). Dissemination bands were then automatically detected in chromosome vs. ARG %GC plots for all ARO-COG groups (Appendix A). Our analysis detected 69,679 ARG homologs in 21,789 complete plasmid sequences, which were assigned to 389 distinct ARO-COG groups.

In recent work, we have shown that genes encoding resistance to the chemotherapeutic agents sulfonamide (*sul*) and trimethoprim (*dfrA*) have been mobilized multiple times from different genomic backgrounds and subsequently disseminated across a wide range of bacterial clades, leading to the observation of multiple dissemination bands in %GC content plots [14]. To ascertain the generality of this phenomenon, we analyzed the number of dissemination bands detected across ARO-COG groups. Our results reveal that dissemination bands are only detected in a relatively small fraction (18.7%) of the 389 ARO-COG groups analyzed (Figure 2, Appendix A), indicating that most mobilized ARG homologs do not disseminate broadly. Among those ARO-COG groups showing evidence of dissemination, most (63.8%) display only a single dissemination band (Figure 3A), suggesting that unique mobilization events are responsible for a majority of broadly disseminated antibiotic resistance genes. Widespread dissemination has been experimentally reported for 29 of the 69 ARO-COG groups with detected dissemination bands, validating the ability of this method to detect ARG dissemination (Appendix A).

Several ARO-COG groups display multiple dissemination bands. We have previously shown that *dfrA* genes (ARO:3001218|COG0262) were disseminated following independent mobilization events in different donor strains [14]. This is reflected in the detection of six different mobilization bands for this ARO-COG group, with putative donor genera %GC ranging from 30% to 53% (Figure 3B). Similarly, four independent dissemination bands are detected in the ARO-COG group for *qnr* genes providing resistance to quinolones (ARO:3000419|COG1357) (Figure 3C), in agreement with previous reports for multiple chromosomal origins for these genes [23,24].

Independent mobilization events have also been postulated as the source of the heavily disseminated CTX-M β-lactamase genes [25]. Even though these mobilization events have been traced to chromosomal genes in different *Kluyvera* species, our analysis detects three distinct dissemination bands corresponding to previously established CTX-M clusters (Figure 3D). In agreement with previous phylogenetic analyses, this suggests that *Kluyvera* species acquired these genes from different donors prior to the mobilization events that fostered their widespread dissemination [25]. Similar results are obtained for OXA β-lactamase genes (ARO:3000017|COG2602), where four dissemination bands are detected. This is consistent with prior reports on their likely divergent chromosomal origins [26].

Among the ARO-COG groups presenting multiple dissemination bands, those conferring resistance to aminoglycosides are significantly overrepresented. Independent dissemination bands are detected for aminoglycoside modifying enzyme classes APH(3′) (ARO:3000126|COG3231) (Figure 3E), ANT(3″) (ARO:3004275|COG1708), AAC(6′) (ARO:3000345|COG1670), APH(6) (ARO:3000151|COG3570) and AAC(3) (ARO:3000322|COG2746), in agreement with previous reports on their wide phylogenetic distribution and postulated multiple independent origins [27,28]. The ARO-COG groups with the largest number of detected dissemination bands correspond to efflux pumps conferring resistance to tetracycline and phenicol (ARO:10002|COG2814; 10 bands) and quaternary ammonium compounds (ARO:10002|COG0477; 8 bands). The %GC plot profiles for these ARO-COG groups reveal extensive mobilization of homologs from these transporter families (Figure 3F). This indicates that efflux pumps conferring resistance to these compounds are present in most chromosomal backgrounds, from which they have been repeatedly disseminated [20,29].

Overall, our analysis indicates that widespread dissemination of ARGs from multiple independent mobilization events is relatively infrequent and restricted to a fairly specific set of ARG families but can be readily detected with %GC plot profiles.

### 2.2. Resistance Mechanisms Determine the Prevalence of ARG Dissemination

The ability to consistently detect dissemination bands in %GC plots for ARO-COG groups led us to develop a Dissemination Index (DI) as a means to systematically quantify and analyze the dissemination of different ARG families. This index measures the proportion of ARG homologs mapping to dissemination bands, with respect to the total number of ARG homologs detected for a given ARO-COG group. As noted earlier, dissemination bands are not detected for most ARO-COG groups, leading to zero values for the dissemination index in 81.3% of ARO-COG groups (Appendix A). For ARO-COG groups with detected dissemination bands, there is no significant correlation between DI values and the number of dissemination bands detected (Pearson *r* = 0.068) (Appendix A). This indicates that the selective pressure underpinning the widespread dissemination of a particular ARG does not necessarily elicit its independent mobilization from different chromosomal backgrounds. This apparently counterintuitive result can be explained by multiple overlapping rationales. For instance, efflux pumps conferring resistance to tetracycline and phenicol are highly mobilized (Figure 3F), leading to repeated dissemination events but also to a low DI due to the large number of non-disseminated ARG homologs detected in plasmids for this ARO-COG group. The high degree of mobilization of efflux pumps may be explained by their cryptic interaction with chromosomal elements coordinating transport processes, which may induce changes in bacterial physiology that promote mobilization [29,30]. On the other hand, genes displaying a single dissemination band, like the NDM β-lactamases (Figure 3A), might have fewer diverse chromosomal sources (not differentiable in a %GC analysis) and may have associated early with highly transmissible MGEs, relieving the pressure for dissemination from independent sources [31].

To identify trends in dissemination index values among the analyzed ARO-COG groups, we analyzed the distribution of DI values when mapping ARO-COG groups to their CARD Resistance Mechanism. The results shown in Figure 4 reveal dramatic differences in DI values for different resistance mechanisms. For obvious reasons, no dissemination is observed for ARG mediating resistance through the absence of the antimicrobial target. Similarly, no dissemination is observed for resistance mediated by reduced permeability to antimicrobials, reflecting both its low prevalence and the fact that in most cases it is achieved through the loss or knockdown mutation of a porin [32]. In contrast, three resistance mechanisms (antibiotic inactivation, target replacement and target protection) display remarkably high DI values (Figure 4). Among these three primary resistance mechanisms, antibiotic inactivation genes are the most consistently disseminated, with an average DI of 81.2%. Several ARO-COG groups encompassing different β-lactamase classes (OKP, TEM, KPC and NDM) and aminoglycoside modifying enzymes (APH(3″) and AAC(6′)-Ib-cr) display DI values in the 99.0–100.0% range, indicating that ARG homologs belonging to these ARO-COG groups are almost exclusively detected in dissemination bands (Appendix A). This remarkable degree of dissemination may be linked to the early adoption of both β-lactams and aminoglycosides in the clinical practice, which led to heightened pressure for the selection of resistant strains [33]. Other highly disseminated ARO-COG groups mediating antibiotic inactivation are the macrolide phosphotransferases (ARO:3000333|COG3173; DI: 97.7%), the fosfomycin thiol-transferases (ARO:3000133|COG2514; 87.5%) and, to a lesser extent, the chloramphenicol acetyltransferases (ARO:3000122|COG0110; DI: 70.3%).

Consistent with recent reports, genes conferring resistance to sulfonamides (ARO:3004238|COG0294) and trimethoprim (ARO:3001218|COG0262) via target replacement also showed high DI values (89.0% and 84.3%, respectively) [14]. Sulfonamides have been used in combination with trimethoprim and other diaminopyrimidines since the 1960’s [33], and genes conferring resistance to both antimicrobial agents are frequently found together on highly transmissible MGEs, such as class 1 integrons and conjugative plasmids [34]. In contrast, alternate variants of the penicillin-binding proteins that confer resistance to methicillin (ARO:3001208|COG0768) showed no evidence of widespread dissemination (Appendix A), as reported previously [35]. Genes *mecA* and *mecC* are typically found as members of the staphylococcal chromosomal cassettes (SCCmec) and known to be mobilizable [36], but the almost exclusive clinical use of methicillin against *Staphylococcus aureus* has likely limited the dissemination of these penicillin-binding proteins to this genus.

Some target protection genes also display relatively large DI values, leading to an average DI of 70.5% for this resistance mechanism. Representative ARO-COG groups for this mechanism include homologs of the *qnr* determinants conferring resistance to fluoroquinolones (ARO:3000419|COG1357; DI: 89.9%), the *tetM* and *tetO* genes providing resistance to tetracyclines (ARO:0000002|COG0480; DI: 78.5%) and the ribosomal protection genes *vgaA* and *vlmR* providing resistance to lincosamide and streptogramin (ARO:3004469|COG0488; DI: 46.5%) (Appendix A). The high dissemination indices of antibiotic inactivation, target replacement and target protection mechanisms are likely associated with their mode of action. These mechanisms confer resistance via highly specific and autonomous processes, providing a rapid and efficient adaptation to the antimicrobial challenge that can be subsequently discarded to maximize reproductive yield [37].

ARGs conferring resistance via target alteration present an average DI of 22.0%, a significantly lower value than those discussed above (Figure 4). Interestingly, however, several target alteration ARGs present large DI values. The colistin-resistance *mcr* genes (ARO:3004465|COG2194) have one of the highest DI values (99.5%), and the *erm* (ARO:3000560|COG0030) and *cfr* genes (ARO:3000202|COG0820), conferring cross-resistance to macrolides–lincosamides–class B streptogramins (MLS_B_) using 23S ribosomal RNA methylation, also display relatively large DI values (62.1% and 71.4%, respectively) (Appendix A). In contrast, many ARG homologs mapping to the target alteration resistance mechanism present very low or zero DI values. Such is the case of homologs of the *ndh* gene conferring resistance to isoniazid (ARO:3003460|COG1252; DI: 19.0%) or the *gyrB* gene conferring resistance to fluoroquinolones (ARO:3000864|COG0187; 0.0%).

The apparently contradictory results for ARO-COG groups mapping to the target alteration mechanism are partly explained by the fact that this CARD Resistance Mechanism category includes both spontaneous mutations to targeted enzymes as well as enzymes that induce chemical modifications in the antimicrobial target protein. The average DI for ARO-COG groups mapping to the target alteration resistance mechanism rises from 22.0% to 38.0% when spontaneous mutants are excluded (Figure 4). This indicates that mutated antimicrobial target variants are rarely disseminated, in contrast to enzymes that mediate the chemical modification of antimicrobial targets. The low dissemination of many ARGs conferring resistance via antimicrobial target alteration, and in particular of mutated variants, may be due to the necessary integration of these altered targets into the host biochemical pathways and physiological processes, which can restrict their ability to confer resistance in unrelated chromosomal backgrounds [38].

Efflux pumps providing resistance to antimicrobial compounds also present a low average dissemination index (30.5%), indicating that they are not generally disseminated (Figure 4). This was also suggested in a recent study, which showed that the chromosomal-to-plasmid ratio for this ARGs is low [20]. Despite this low ratio, our results reveal that the number of plasmid-borne antibiotic efflux ARG homologs vastly outnumbers the number of ARG homologs observed for any other resistance mechanism (Appendix A). Only a few ARO-COG groups mapping to antibiotic efflux, like the one conferring resistance to tetracycline and phenicol, show evidence of widespread dissemination (Figure 3F). A few efflux pumps display very high DI values, indicating that they are predominantly disseminated and not frequently mobilized from taxonomically close chromosomal backgrounds. Some variants of the *qac* genes (ARO:10002|COG2076; DI: 98.8%) are among the most notable of these instances (Appendix A). These efflux pumps are integron-borne and confer resistance to quaternary ammonium agents. Their high dissemination index value is driven by the most prevalent variant, *qacEΔ1*, which is typically found in the widespread class 1 integrons with *sul* and *dfr* genes [39].

Overall, our results support a general framework wherein the degree of dissemination of ARGs is proportional to their mutational accessibility and their availability in the bacterial pangenome. Intuitively, specialized enzymes capable of selectively inactivating antimicrobials or of protecting or replacing antimicrobial targets without impacting on their function cannot be readily evolved and must be acquired via HGT. This leads to high dissemination indices that reflect their widespread dissemination across multiple bacterial clades. In contrast, the ability to repurpose existing efflux pumps or to generate viable resistant target variants through mutation implies that chromosomal sources for resistance genes are likely available within any given bacterial clade. This leads to high rates of mobilization, but simultaneously lessens the selective pressure for the widespread dissemination of these resistance genes.

### 2.3. Antimicrobial Agent Classes Present Different Dissemination Patterns

The analysis of %GC content to monitor ARG dissemination also reveals different patterns of dissemination as a function of the antimicrobial agent they confer resistance to (Figure 5; Appendix A). Among antimicrobial classes, the different patterns observed reflect the availability and degree of dissemination of different resistance mechanisms for any given drug class. Sulfonamides (Figure 6A), for instance, present an aggregate DI of 66.1%. ARG homologs conferring resistance to these chemotherapeutic agents are dominated by highly disseminated *sul* genes’ mediating target replacement, which have been heavily disseminated through their association with class 1 integrons [15]. The highly disseminated *sul* genes are sometimes annotated in CARD as specific for sulfonamides, but more frequently as having multiple targets (sulfonamides and sulfones), yielding DI values of 94.7% and 88.8%, respectively. Sulfonamide resistance can also be conferred by nonspecific efflux pumps, but these ARG homologs present very low dissemination (3.8%), driving down the aggregate DI value for this drug class.

Other drug classes, such as the β-lactams (Figure 6B; aggregate DI: 43.9%) present profiles similar to the one observed for sulfonamides. For β-lactams, heavily disseminated β-lactamases mediating specific antibiotic inactivation present high DI values (75.6%), but the overall dissemination index for this antibiotic class is impacted by the abundance of nonspecific, infrequently disseminated efflux mechanisms also targeting β-lactams. The contribution of non-disseminated efflux pumps to the overall plasmid resistome dissemination profile is even larger for other antimicrobial classes, such as the fluoroquinolones (Figure 6C; aggregate DI: 29.7%) or the macrolides (Figure 6D; aggregate DI: 33.5%), where multiple resistance mechanisms are detected. In these cases, ARGs linked to specific resistance mechanisms (antibiotic inactivation and target protection, respectively) display very high dissemination indices when compared to nonspecific mechanisms. This is particularly true for nonspecific antibiotic efflux mechanisms conferring cross-resistance to these antimicrobials. Homologs for these efflux pumps are detected abundantly in the plasmidome, but present low levels of dissemination in agreement with recent reports [20].

Several antimicrobial classes display a remarkable absence of homologs mediating resistance by either efflux pumps or specific mechanisms. The streptogramins, for instance, show substantial dissemination of nonspecific ARG homologs mapping to target alteration (DI: 63.4%) and protection mechanisms (DI: 45.8%) (Figure 6E). These ARG homologs mediate, respectively, targeted methylation of ribosomal subunits [40,41] and ribosomal protection [42], conferring resistance to multiple antimicrobial classes including lincosamides, oxazolidinones, phenicol and streptogramins [43]. In contrast, a substantial proportion of detected ARG homologs conferring resistance to tetracyclines map to highly disseminated specific efflux pumps and their transcriptional regulators (DI: 61.9%), as well as nonspecific efflux pumps with low dissemination index (DI: 19.5%). For tetracyclines, specific target alteration (DI: 78.1%) and protection (DI: 50.1%) mechanisms are highly disseminated but appear to play a secondary role in terms of the amount of ARG homologs detected in the plasmidome (Figure 6F).

Interestingly, several antimicrobial classes display low or negligible dissemination profiles (Figure 5). For some of these antimicrobials, the most likely explanation for their low dissemination index is their organism-specific usage, typically restricted to the clinical setting. This is the case of isoniazid, ethionamide and prothionamide, which are almost exclusively used to treat *Mycobacterium tuberculosis* infections [44]. For other antimicrobial classes, the observed low DI values are likely associated with resistance being mediated almost exclusively by nonspecific efflux pumps (e.g., antibacterial fatty acids, aminocoumarin or bicyclomycin) or by the mutational accessibility of variants conferring resistance (e.g., cycloserine or polyamine antibiotics) [45].

The different patterns of dissemination observed for antimicrobial classes may reflect different evolutionary processes. Exposure to high doses of an antimicrobial agent can promote incorporation via HGT of specific resistance genes that provide an immediate evolutionary advantage [46]. These ARGs often act through largely autonomous mechanisms, such as antibiotic inactivation, and target specific drug classes. In contrast, ARGs providing cross-resistance often operate through nonspecific mechanisms such as efflux pumps. These ARGs are frequently encoded by the bacterial chromosome and operate in coordination with other physiological processes. This facilitates their mobilization, even in the absence of selective pressure, but may limit their widespread dissemination to unrelated chromosomal backgrounds [20,29,30].

### 2.4. Conjugative Plasmids Mediate Dissemination of Antimicrobial Resistance Genes

To study the contribution of different plasmid classes on plasmid-borne antimicrobial resistance dissemination, we used the MOB_suite to predict plasmid mobility types. We then constructed a bipartite network using Gephi with the predicted ARGs and PLSDB plasmids as nodes. The presence of ARG homologs in plasmids was depicted as network connections and node coloring accounted for ARO-COG dissemination indices and predicted plasmid mobility class, respectively. In accordance with previous results, our analysis shows that conjugation is the main dissemination driver of most ARGs [1], providing independent validation to the proposed DI as an estimator of ARG dissemination.

The association pattern of ARO-COG groups and plasmids observed in the network (Figure 7, Appendix A) reveals that non-disseminated gene families (DI = 0) are more common in non-mobilizable plasmids (44.3%) than in mobilizable (33.7%) and conjugative plasmids (22.0%). In contrast, ARO-COG groups with very large dissemination indices (DI > 75%) are mainly associated with conjugative plasmids (69.4%). The rest of ARO-COG groups showing evidence of dissemination (0% < DI < 75%) are also more likely to be found in non-mobilizable plasmids (38.4%) than in conjugative (34.0%) and mobilizable plasmids (27.6%). The latter case is of interest because ARO-COG groups with evidence of dissemination appear to be more common in non-mobilizable than in mobilizable plasmids. This suggests that conjugation mediated by helper *tra* genes might not play a fundamental role in the widespread dissemination of ARG in natural communities.

Differences in chromosomal %GC content are a well-established proxy for phylogenetic divergence, which is known to limit the efficiency of conjugation [21,47]. Our results illustrate that the %GC content of many ARG homologs is effectively independent of the %GC of the host on which the plasmid harboring them has been isolated, providing the means to identify highly disseminated ARGs (Figure 1). The observed %GC independence profile of highly disseminated ARGs can be explained by two complementary processes. On the one hand, selective pressure driven by exposure to antimicrobials could overcome phylogenetic barriers to plasmid conjugation, facilitating the proliferation of the ARG-harboring plasmid on a nonnative host. On the other hand, following an initial conjugation event from a nonnative plasmid harboring ARGs in integrons or other MGEs, the same selective pressure might promote the transmission of the ARGs to a plasmid compatible with the host, facilitating its subsequent spread. Analysis of plasmid %GC content relative to chromosomal host %GC content reveals that plasmid and chromosome %GC content are strongly correlated (Pearson *r* = 0.893) when all ARG homologs are considered. This robust correlation is maintained (Pearson *r* = 0.792) when only plasmids harboring ARG homologs in dissemination bands are considered (Appendix A). This suggests that widespread dissemination of ARGs is predominantly mediated by transmission of ARG homologs to plasmids compatible with their hosts and highlights the importance of integrons and other plasmid-borne MGEs in the spread of antimicrobial resistance [48,49].

## 3. Materials and Methods

### 3.1. Antimicrobial Resistance Gene Prediction, Classification and Plasmid Analysis

Putative antimicrobial resistance genes (ARGs) were predicted on plasmid sequences from the PLSDB plasmid database [50] using all available protein sequences from the Comprehensive Antibiotic Resistance Database (CARD) collection as queries [51]. The PLSDB complete plasmid assemblies and CARD protein sequences were downloaded from the NCBI Nucleotide and Protein databases in FASTA protein format. Homologs of CARD protein sequences in the PLSDB were detected with BLASTP using an e-value <1 × 10^−20^ and a query coverage >75% as limiting thresholds [52]. Each putative ARG identified in PLSDB was linked to a CARD antimicrobial resistance (AMR) Gene Family, Drug Class and Resistance Mechanism using the accessory information from the CARD query with the lowest e-value in the BLASTP analysis. CARD annotation on genes conferring resistance through mutation was compiled and assigned to corresponding homologs. Plasmid type and host information for each putative ARG were obtained from the PLSDB. Predicted ARGs were mapped to Clusters of Orthologous Groups (COG) [53] using HMMER (hmmscan) [54] with a limiting e-value of 1 × 10^−5^ and otherwise default parameters. Putative ARGs not mapping to any COG were removed from the analysis. Predicted ARGs were then grouped based on their CARD AMR Gene Family and their mapping COG. These groups were assigned compound identifiers using the Antibiotic Resistance Ontology (ARO) accession for the AMR Gene Family and the COG accession and are henceforth referred to by their ARO-COG accessions.

### 3.2. Antimicrobial Resistance Gene Dissemination Analysis

ARG dissemination analysis is based on the study of correlation between the %GC of ARG homologs identified in plasmids and the chromosomal %GC content of the plasmid isolation hosts. For widely disseminated genes, it is expected that ARG %GC will be independent of host chromosomal %GC, displaying dissemination bands that can be detected via computational analysis (Figure 1). For each ARO-COG group with more than 10 ARG homologs mapping to it, protein and coding DNA sequences for the ARG homologs identified in PLSDB were downloaded from the NCBI GenBank database for %GC content analysis. %GC content for ARG homologs was computed using BioPython [55]. The host %GC content for each predicted ARG was determined by matching the reported plasmid host strain, species or genus against a RefSeq representative complete genome from the NCBI Assembly database. The correlation between the %GC content of the ARGs and that of their host genome was depicted as a scatter plot using a custom Python script. Redundant and nonredundant sets of antibiotic resistance coding sequences were generated by clustering all ARG homologs identified in PLSDB using Usearch with a 90% similarity threshold and otherwise default parameters [56].

For the systematic analysis of the dissemination of antimicrobial resistance genes, we computed the Dissemination Index (DI) for each ARO-COG group. The DI is defined as the fraction of redundant ARG homologs in the ARO-COG group that map to dissemination bands, relative to the total number of ARG homologs in the ARO-COG group. Dissemination bands were calculated using a custom algorithm inspired by the gapless extension phase of BLAST [52] (Appendix A). Briefly, for a given scatter plot contrasting host (*x*-axis) and redundant ARG (*y*-axis) %GC content, the algorithm identifies a dissemination band as any segment of ARG homologs spanning more than a detection span *DS* (15% by default) in the *x*-axis (i.e., sets of ARG homologs distributed among hosts with a %GC divergence greater than 15%). Scanning proceeds by 1% increments on the *y*-axis until a dissemination band meeting the condition above is detected. Upon detecting a dissemination band, a bidirectional extension phase ensues. The band *x*-axis span is saved as the current span, and subsequent 1% increments on the *y*-axis are incorporated to the band if their span is at the most *d* (5 by default) percentual points smaller than the current span. Otherwise, extension of the dissemination band in that direction is terminated. To limit the occurrence of false positives, the average number of ARG homologs in any given 1% increment containing data is computed, and only dissemination bands exceeding this expected value are considered for analysis. To compute DI values for CARD metadata-based groupings, such as Resistance Mechanism or Drug Class, the CARD metadata and dissemination band status associated with each predicted ARG homolog is tabulated independently to generate a DI value for the entire metadata grouping.

The scripts used for data collection and analysis are available at the GitHub ErillLab repository (https://github.com/ErillLab/, accessed on 14 December 2022).

## 4. Conclusions

The ability to assess and monitor ARG dissemination is critical to inform antibiotic policy and develop predictive models of resistance dissemination. This work shows that analysis of the correlation between ARG and plasmid-host %GC content provides the means to efficiently leverage publicly available plasmid sequence data to quantitatively assess ARG dissemination with a visually interpretable and easily scalable dissemination index (DI). Furthermore, our analysis pipeline can be generalized to study other genetic elements relevant to antimicrobial resistance and microbial adaptation, such as siderophores or genes determining plasmid incompatibility. Results from the %GC content analysis show that ARG dissemination is driven predominantly by resistance mechanism. This finding confirms previous results and provides insights into the selective pressures underpinning ARG dissemination. Our analysis also reveals that most broadly disseminated genes originated with a single mobilization event, and that their dissemination has been mediated primarily by conjugative plasmids. Our data on ARG dissemination reveals substantial differences among antimicrobial classes, reflecting in some cases antibiotic policy. Overall, our findings illustrate how a fast and intuitive analysis can be used to reveal fundamental aspects of antimicrobial resistance dissemination and provide the basis for subsequent experimental work to gauge the impact of disseminated ARGs on antibiotic resistance in natural and clinical isolates.

## Figures and Tables

**Figure 1 antibiotics-12-00281-f001:**
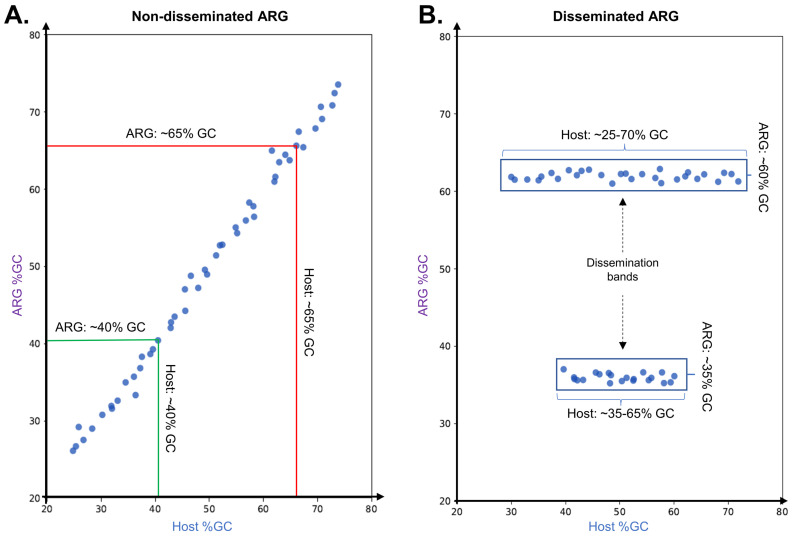
Graphical representation of hypothetical scenarios for the correlation between the %GC content of plasmid-borne ARG and that of their host genome. (**A**) The ARG %GC content is expected to align with the %GC content of the host chromosome when a resistance gene has been mobilized in multiple lineages but has been transferred only among closely related species. (**B**) The ARG %GC content is essentially independent of the host %GC content when a resistance gene has been rapidly disseminated from one or more mobilized instances to a heterogenous group of hosts, generating visible “dissemination bands”.

**Figure 2 antibiotics-12-00281-f002:**
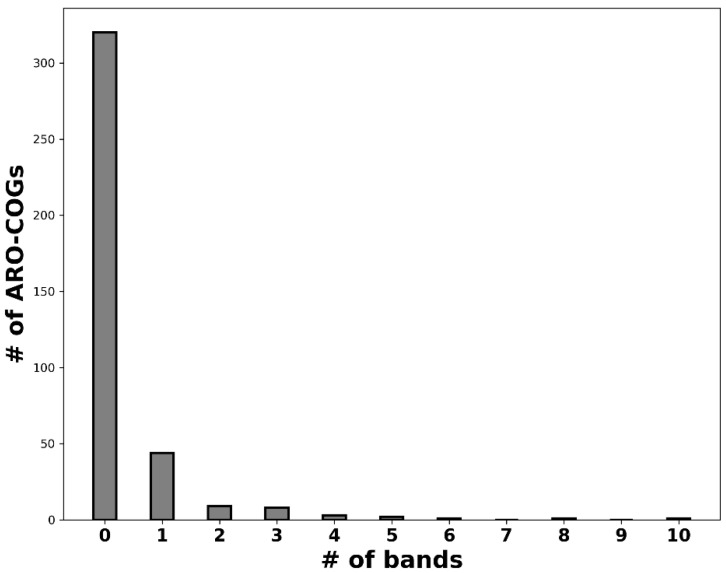
Histogram depicting the distribution of the number of predicted dissemination bands per ARO-COG group.

**Figure 3 antibiotics-12-00281-f003:**
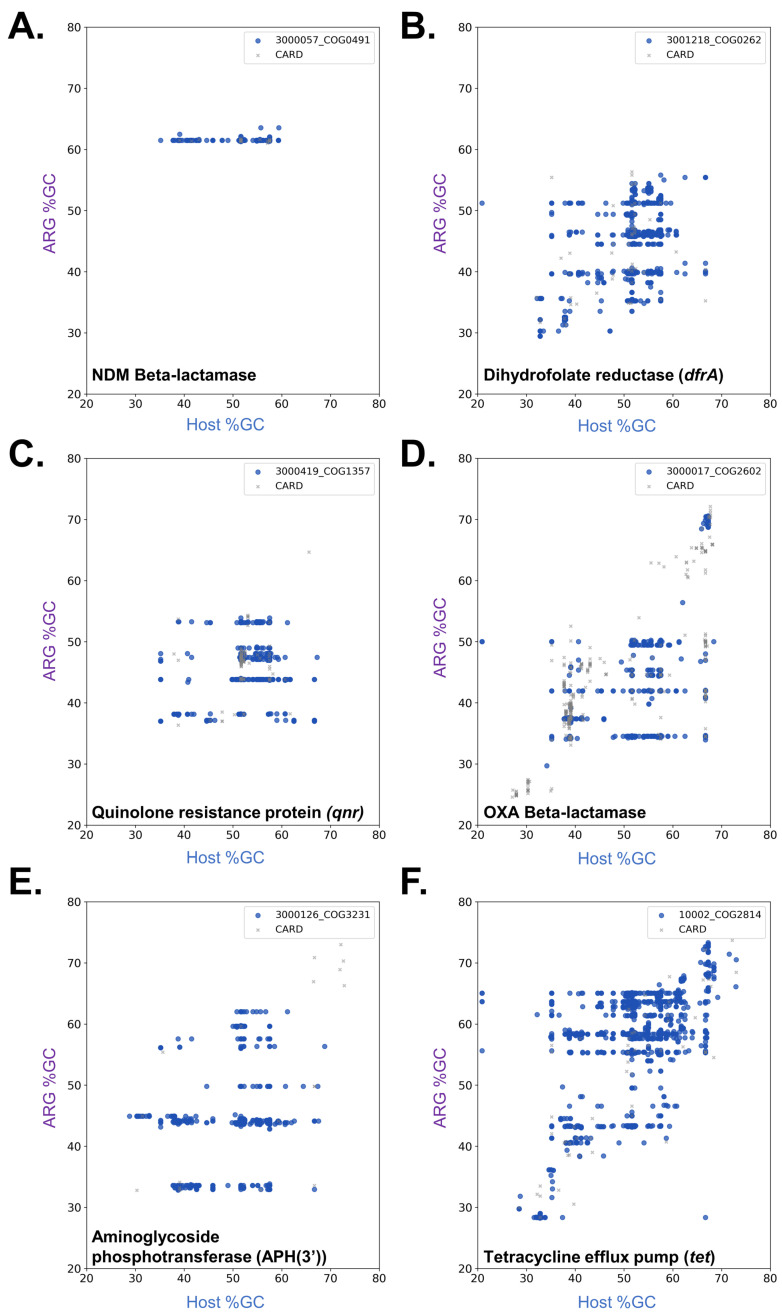
Scatter plots showing the correlation between the %GC content of mobile NDM beta-lactamase (**A**), *dfrA* (**B**), *qnr* (**C**), OXA beta-lactamase (**D**), APH(3′) (**E**) and *tet* (**F**) genes and that of their host chromosome. Only genes encoding redundant protein homologs (>90% sequence similarity) are displayed. Experimentally validated resistance genes annotated in CARD that map to each ARO-COG group are shown as grey crosses.

**Figure 4 antibiotics-12-00281-f004:**
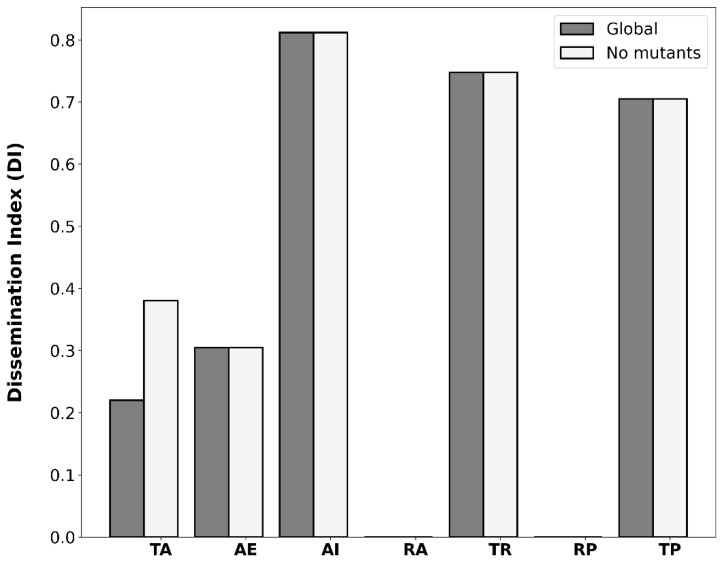
Bar plot of DI values aggregated on CARD Resistance Mechanism: target alteration (TA), antibiotic efflux (AE), antibiotic inactivation (AI), resistance by absence (RA), target replacement (TR), reduced permeability (RP) or target protection (TP). The dissemination status (in band, not in band) of each ARG homolog mapping to a given resistance mechanism is used for the computation of the aggregate DI value. The DI values of each resistance mechanism when removing any ARG homologs annotated as conferring resistance via mutation are also reported.

**Figure 5 antibiotics-12-00281-f005:**
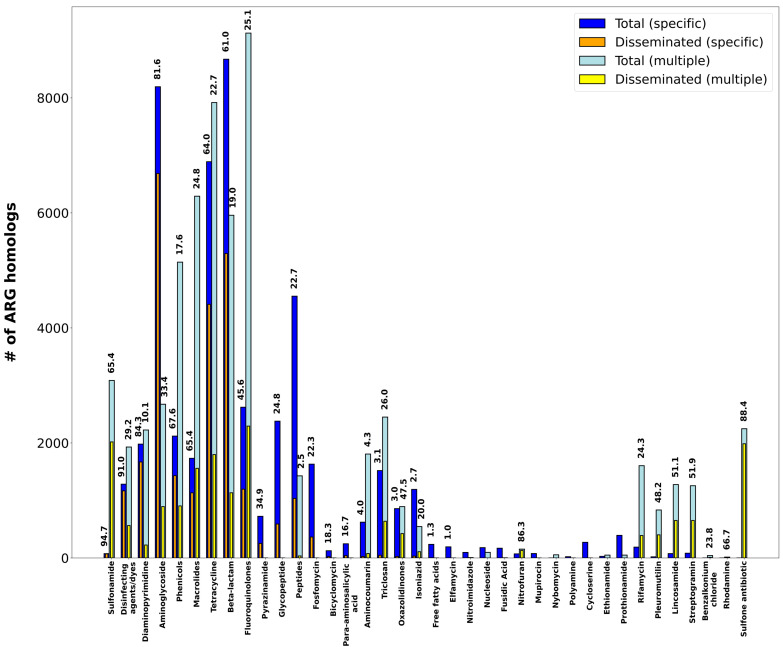
Bar plot showing the total vs. disseminated number of ARG homologs aggregated by CARD Drug Class. “Specific” denotes DI values computed solely on ARG homologs matching a single CARD Drug Class. “Multiple” denotes DI values computed solely on ARG homologs matching several CARD Drug Classes. Dissemination index values for each group are superimposed when larger than 0%.

**Figure 6 antibiotics-12-00281-f006:**
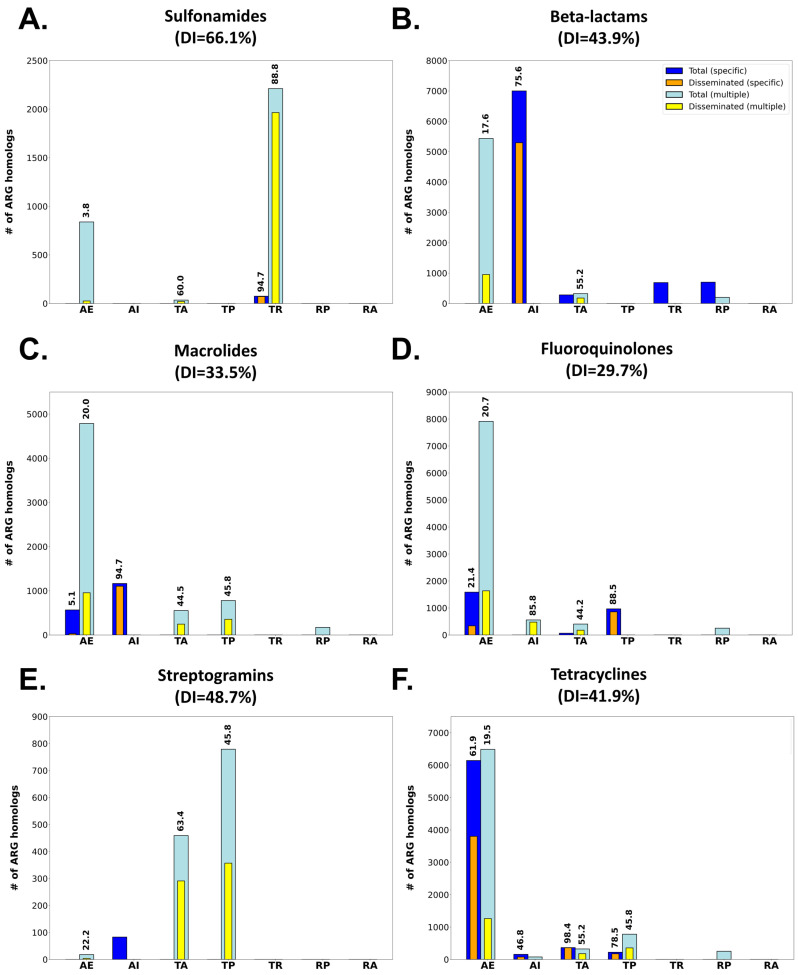
Bar plots showing the total vs. disseminated number of ARG homologs matching single (Specific) or several (Multiple) CARD Drug Classes and conferring resistance to sulfonamides (**A**), beta-lactams (**B**), macrolides (**C**) fluoroquinolones (**D**), streptogramins (**E**) or tetracyclines (**F**). ARG homologs are also split according to their CARD Resistance Mechanism: target alteration (TA), antibiotic efflux (AE), antibiotic inactivation (AI), resistance by absence (RA), target replacement (TR), reduced permeability (RP) or target protection (TP). Dissemination index values for each group are superimposed if larger than 0%.

**Figure 7 antibiotics-12-00281-f007:**
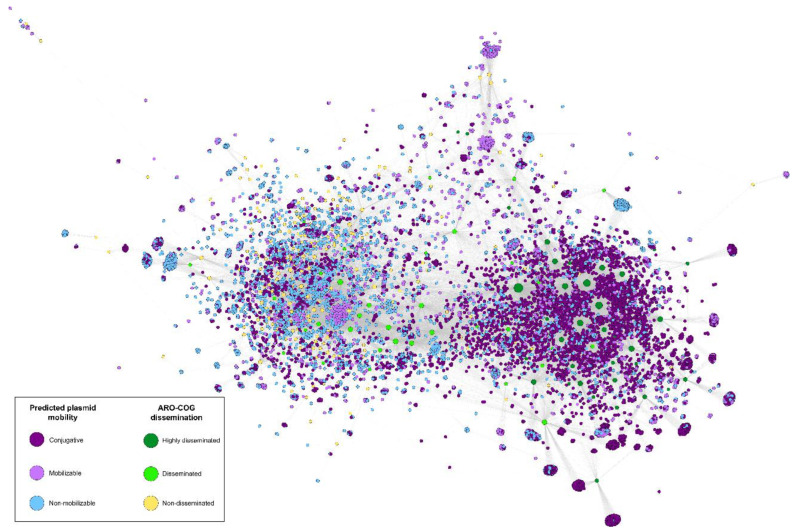
Bipartite network illustrating the correlation between predicted plasmid mobility and ARG dissemination. Plasmid mobility was predicted using the MOB_suite. “Highly disseminated” denotes ARO-COGs with DI > 75%, “Disseminated” stands for ARO-COGs with 0% > DI > 75% and “Non-disseminated” for ARO-COGs with DI = 0%. An interactive version of the network is available at https://miquelsanchezosuna.github.io/arg_gc_network/ (accessed on 14 December 2022).

## Data Availability

Nucleotide and protein sequences analyzed in this study have been downloaded from publicly available National Center for Biotechnology Information databases. The scripts used for data collection and analysis can be obtained at the GitHub ErillLab repository (https://github.com/ErillLab/, accessed on 14 December 2022).

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
