# Peer review of "Systematic In Silico Assessment of Antimicrobial Resistance Dissemination across the Global Plasmidome"

_antibiotics, 2023, doi:10.3390/antibiotics12020281_

Round 1

Reviewer 1 Report

The first paragraph of the introduction is too general, since the focus of the study is bacterial resistance this part should be specific on bacteria. This extensive work seeks to use plasmid marking to trace the mechanism of resistance gene distribution in bacteria. I feel the manuscript is well written and the research is well conducted. I will suggest some spelling  and grammar checks for the manuscript

Author Response

The first paragraph of the introduction is too general, since the focus of the study is bacterial resistance this part should be specific on bacteria. This extensive work seeks to use plasmid marking to trace the mechanism of resistance gene distribution in bacteria. I feel the manuscript is well written and the research is well conducted. I will suggest some spelling  and grammar checks for the manuscript

We have updated the first paragraph of the introduction to make it specific for bacteria. We have also revised spelling and grammatical issues in the manuscript with the assistance of a native English speaker.

Reviewer 2 Report

The work is profuse, original, well articulated and designed, I think it reaches the quality of the magazine, however there are several comments that I attach that should be considered.

In lines 17-19 “Analyzing the %GC content of plasmid-borne antimicrobial resistance genes versus their host genome %GC content provides the means to efficiently detect and quantify dissemination of antimicrobial resistance gene”

It is an interesting approach since the %GC is a very generic and broad a priori value in the sense that it does not provide specific information on the metabolism of the bacteria or adaptive strategies. Perhaps it would be interesting to correlate the %GC of a plasmid with its recombination potential (particularly homologous recombination), since this factor could be conditioning the efficiency of horizontal gene transfer phenomena.

In this work we automate %GC content analysis to perform a comprehensive analysis of the plasmid-encoded resistome.

Referring to this sentence, it would be necessary to emphasize that the resistome analysis is carried out on those genes for which there is already reference in databases that participate in the process of resistance to antibiotics, leaving genes present in the plasmids that could help through a cryptic role. Which I consider to be of special relevance to be able to infer conclusions regarding their mobility potential (independent/dependent)

We find that the mechanism by which antimicrobial resistance is conferred is the primary determinant of antimicrobial resistance gene dissemination, and that genes conferring resistance to specific antimicrobial classes are substantially more disseminated than those providing broad-spectrum resistance

From my perspective, this point is the most relevant element of the work, however, I believe that the discussion should go deeper into this aspect, and this aspect is not perceived to be of special relevance. From my point of view, the phenomena of dissemination of genes associated with resistance should have a certain correlation with the intensity and type of selection pressure where the donor organism lives. At work it seems to be focused independently, and it seems new to me, however I understand that this type of result could be supported by other works that do not appear in the discussion. Regarding the non-specific resistance mechanisms, we might think that they should have a greater capacity for intrinsic dissemination in the absence of selection pressure, (this particular aspect was not considered in the work and at least it would be necessary to justify it with the scientific bibliography) while the HGT processes increase in intensity under specific conditions of selection pressure, as has already been reported in highly relevant works (Martínez, J., Coque, T. & Baquero, F. Prioritizing risks of antibiotic resistance genes in all metagenomes. Nat Rev Microbiol 13, 396 (2015). https://doi.org/10.1038/nrmicro3399-c2).

In lines 50, 51 and 52, it says

 “In these environments, bacteria can leverage two alternative processes to mitigate the effects of antimicrobial agents. On the one hand, mutations in genes typically associated with the mechanism of action of the drug may occur and be selected for.”

In this sense, it is necessary to add that one of the mechanisms to mitigate the effect of antimicrobial agents does not have to be directly associated with genes with an antimicrobial effect, it can be indirectly associated, regulating expression, or acting as a redundant element in the network. genomics or act in cryptic ways.

When the sequence of the plasmids was analyzed, was it done completely?, or was it only restricted to sequences associated with resistance?

Would it be possible to think of HGT phenomena of sequences that have been configured with evolutionary convergence and not by dissemination? In this sense, it would be worth assuming that there is a high variability in the sequences of the proteins associated with resistance in areas not related to the catalytic site, how could one distinguish one type of sequence from another?

A direct consequence of the differences in the %GC of one species with respect to another could be associated with the amount of energy allocated to the replication process, due to a higher density of hydrogen bonds per unit length of sequence, it could have Does this metabolic effect of energy flow have any effect in reference to the results obtained?

How do you explain that the transfer of conjugative plasmids that requires specific recognition by the donor and recipient is independent of the %GC? Wouldn't it be more relevant to focus on the determinants that allow plasmid transfer than %GC?

In lines 105-1028, it says

“In addition, our results show that ARG conferring resistance to specific  antimicrobial classes present higher dissemination than genes conferring cross-resistance to multiple antibiotics. We report that different antimicrobial classes present remarkably diverse ARG dissemination profiles, and that plasmid-based ARG dissemination is not detected for many restricted-use antimicrobials.”

I understand that in this paragraph it would be worth clarifying that a specific dissemination event should be associated with the intensity of the selection pressure on the one hand and the resistance phenotype of the bacteria, since it would make evolutionary sense to promote a survival result. in the short term. On the other hand, from an evolutionary perspective it is logical to think that the adaptation mechanisms to multiple antimicrobials, or non-specific resistance mechanisms should be housed in the bacterial chromosome, which ensures greater stability over time and a vertical transfer assuming that it is information. genetics that have greater versatility and ecosystem functionality. I understand that this thesis, which is implicit in your reasoning, should be explicitly and adequately referenced. This thesis is later corroborated in the document, but is not referenced by “This indicates that efflux pumps conferring resistance to these compounds are present in most chromosomal backgrounds, from which they have been repeatedly disseminated.”

You indicate that the Antibiotic resistance Ontology (ARO) was used in other works, arguing that it is a technique that restricts the analysis because it reduces the problem and makes it discrete, while your method incorporates a greater amount of data, however from my point of view Given the %GC can be associated with a huge amount of genetic information. I think that what would be interesting in this case would be to have focused his analysis on the GC content of the compatibility factors in the HGT phenomena associated with conjugative plasmids. I think it is a valid analysis, but very general. I believe that a paragraph should be included to clarify this point.

In the process of analyzing the sequences, I understand that there should be a difference between the sequences from bacteria that inhabit a wide diversity of ecological niches with respect to those bacteria circumscribed to much more specific ecosystem conditions (for example, sequences from extremophile bacteria, where the ecosystem conditions are much more stable), correlative analyzes between these two major categories of content would allow ratifying his thesis or explaining the generality of this phenomenon. In this sense, in their analysis they refer to Kleuverya, but it is a very widespread bacterium from the point of view of the variety of environmental factors to which it adapts. In baceria with less susceptibility to accept mobile elements foreign to the ecosystem, such as extremophile bacteria, would the specific elements associated with the transfer behave in the same way? Would they be determined by the %GC?

In another part of the document he says;

This indicates that the selective pressure underpinning the widespread dissemination of a particular ARG does not necessarily elicit its independent mobilization from different chromosomal backgrounds. This apparently counterintuitive result can be explained by multiple over- lapping rationales

In reference to this assertion it is convenient to consider the following, in the work it is assumed that the adaptive strategy of bacteria to face the presence of antibiotics is based on processes of independent dissemination of mobile elements that carry ARG, and for this it exemplifies a situation in where this phenomenon is not observed, but can be explained by overlapping. However, it is logical to think that bacterial genomic networks (gene regulatory network GRN or “chromosomal backgrounds”), by functioning in an intrinsically cooperative way in free-living bacteria and in those that are forming microcolonies (which is the ecosystem context where it is most favors HGT), modify their interactome after having integrated a mobile genetic element, resulting with greater or lesser efficiency in the adaptive process. In this sense, it would be worth considering that the phenomena of regulation of expression or regulation of transport (for example, porin-associated permeability, MFS or RND) is under the control of the bacterial chromosome, therefore there must be an interaction phenomenon between the pre-existing information in the bacterium and that of the conjugative plasmid that was integrated, generating an adaptive result that "cryptic" way (that is, through processes of regulation of expression or energy balances in metabolic pathways) generates a response in bacterial cell physiology that promotes an increase in the rate of mobilization. In this sense, it would be worth using this same methodology to determine the mobilization intensity of a siderophore for heavy metals, for example, and to determine its dispersion in a medium with a selection pressure imposed by the presence of antibiotics, for example. From my point of view, the modification of the interactome of the bacterium causes this process to be perceived as independent when it is not, but rather is induced by a cryptic mechanism.

In contrast, three resistance mechanisms (antibiotic inactivation, target replacement and target protection)  display remarkably high DI values (Figure 4)

In this statement, I believe it is convenient to emphasize that the mechanisms where a high rate of dissemination is perceived are specific processes, and therefore more energy efficient to adapt to the presence of antimicrobials. From an adaptive perspective, it is useful information for a reasonably short time, and therefore more susceptible to being used by bacteria through HGT phenomena that makes it possible, over generations, to eliminate said information from the genetic background, improving with they reproductive yields. In this sense, I think it is convenient to include this type of information with a bibliographical citation.

Elsewhere in the document you can read;

ARGs conferring resistance via target alteration present an average DI of 22.0%, a 272 significantly lower value than those discussed above

Referring to this result, it would be worth taking up the comment that I previously wrote in reference to the interactome, in this sense, the modification of a target requires a route from the receiving bacterium to integrate the information present in the plasmid, (for example a transcription factor, even the lack of a biochemical pathway), is not an autonomous or independent process within cell physiology, this determines that the elements that induce or promote dissemination require an element or specificity factor associated with the transfer in reference to the physiology of donor/recipient bacteria

Finally, it says in the document:

The apparently contradictory results for ARO-COG groups mapping to the target alteration mechanism are explained by the fact that this CARD Resistance Mechanism category includes both spontaneous mutations to targeted enzymes as well as enzymes that induce chemical modifications in the antimicrobial target protein

In this statement, I do not consider that the mutation rate is a factor that explains this result; Indeed, spontaneous mutations do occur, but what modifies the mutations are the protein-protein interaction network, which can be more or less efficient. This situation is different when the interaction does not occur, because the biochemical pathway is missing, or because the protein or enzyme that "chains" the resistance mechanism is not present. In other words, the capacity or potential for information capture through conjugative plasmids must necessarily be regulated by the receiving bacterium, in my opinion due to the potential combinatorial capacity of the plasmid information with the bacterial GRN.

Author Response

The work is profuse, original, well articulated and designed, I think it reaches the quality of the magazine, however there are several comments that I attach that should be considered.

In lines 17-19 “Analyzing the %GC content of plasmid-borne antimicrobial resistance genes versus their host genome %GC content provides the means to efficiently detect and quantify dissemination of antimicrobial resistance gene”

It is an interesting approach since the %GC is a very generic and broad a priori value in the sense that it does not provide specific information on the metabolism of the bacteria or adaptive strategies. Perhaps it would be interesting to correlate the %GC of a plasmid with its recombination potential (particularly homologous recombination), since this factor could be conditioning the efficiency of horizontal gene transfer phenomena.

Our study does not analyze plasmid %GC content, but focuses instead on analyzing the relative difference between the %GC content of plasmid-borne antimicrobial resistance genes and chromosomal %GC content of the plasmid isolation hosts. At this point we are not aware of a reliable software pipeline to broadly assess plasmid recombination potential, but we agree with the reviewer that studying how the recombination potential of plasmids correlates with plasmid %GC content would be an interesting avenue for future research.

In this work we automate %GC content analysis to perform a comprehensive analysis of the plasmid-encoded resistome.

Referring to this sentence, it would be necessary to emphasize that the resistome analysis is carried out on those genes for which there is already reference in databases that participate in the process of resistance to antibiotics, leaving genes present in the plasmids that could help through a cryptic role. Which I consider to be of special relevance to be able to infer conclusions regarding their mobility potential (independent/dependent)

Our the study is limited to antimicrobial resistance genes annotated in the CARD database, and we agree with the reviewer that this should be stated explicitly, since other plasmid-encoded genes could indirectly facilitate or directly provide resistance to antimicrobials through cryptic roles, and hence have significant impact on the resistance profile of plasmids. We have amended the corresponding abstract sentence to indicate this (Lines 20-21), and we have elaborated on the role of cryptic resistance genes in the text (Lines 57-58, 232-235).

We find that the mechanism by which antimicrobial resistance is conferred is the primary determinant of antimicrobial resistance gene dissemination, and that genes conferring resistance to specific antimicrobial classes are substantially more disseminated than those providing broad-spectrum resistance

From my perspective, this point is the most relevant element of the work, however, I believe that the discussion should go deeper into this aspect, and this aspect is not perceived to be of special relevance. From my point of view, the phenomena of dissemination of genes associated with resistance should have a certain correlation with the intensity and type of selection pressure where the donor organism lives. At work it seems to be focused independently, and it seems new to me, however I understand that this type of result could be supported by other works that do not appear in the discussion. Regarding the non-specific resistance mechanisms, we might think that they should have a greater capacity for intrinsic dissemination in the absence of selection pressure, (this particular aspect was not considered in the work and at least it would be necessary to justify it with the scientific bibliography) while the HGT processes increase in intensity under specific conditions of selection pressure, as has already been reported in highly relevant works (Martínez, J., Coque, T. & Baquero, F. Prioritizing risks of antibiotic resistance genes in all metagenomes. Nat Rev Microbiol 13, 396 (2015). https://doi.org/10.1038/nrmicro3399-c2).

We agree with the reviewer that the observed difference in dissemination between antimicrobial class specific and non-specific ARGs is an important result of this work. We have added these elements to the discussion and provided references to substantiate the main points (Lines 408-418).

In lines 50, 51 and 52, it says

 “In these environments, bacteria can leverage two alternative processes to mitigate the effects of antimicrobial agents. On the one hand, mutations in genes typically associated with the mechanism of action of the drug may occur and be selected for.”

In this sense, it is necessary to add that one of the mechanisms to mitigate the effect of antimicrobial agents does not have to be directly associated with genes with an antimicrobial effect, it can be indirectly associated, regulating expression, or acting as a redundant element in the network. genomics or act in cryptic ways.

We agree with the reviewer that this is an important point to make and we have incorporated text to reflect it (Lines 56-58).

When the sequence of the plasmids was analyzed, was it done completely?, or was it only restricted to sequences associated with resistance?

We only analyzed homologs of genes present in the CARD database, as stated in the methods. We used the plasmid sequence only to perform plasmid mobilization typing with MOB_suite.

Would it be possible to think of HGT phenomena of sequences that have been configured with evolutionary convergence and not by dissemination? In this sense, it would be worth assuming that there is a high variability in the sequences of the proteins associated with resistance in areas not related to the catalytic site, how could one distinguish one type of sequence from another?

By definition, convergent evolution leads to analogous phenotypes with no underlying sequence similarity (given the independent genetic origin of the trait). For instance, several ribosomal modification genes (e.g. erm, cfr) confer resistance to lincosamides, but no significant sequence similarity.

Our analysis shows that in some cases, genes have disseminated from markedly different chromosomal backgrounds (multiple bands). One would expect that these sequences, while homologous, present substantial sequence divergence, but higher sequence similarity within catalytic sites. This can be illustrated by MSA in representative cases. For instance, different classes of the sul gene conferring resistance to sulfonamides present substantial sequence divergence (Sanchez-Osuna et al. 2019), but show high conservation on their modified active site, which has just been recently characterized (https://www.biorxiv.org/content/10.1101/2022.06.30.498311v1). The same reasoning applies to other antimicrobial resistance genes, such as the Cfr/RlmN methyltransferases, which display a conserved CX3CX2C in spite of substantial sequence divergence in other parts of the sequence (https://doi.org/10.1093/nar/gkp1142).

A direct consequence of the differences in the %GC of one species with respect to another could be associated with the amount of energy allocated to the replication process, due to a higher density of hydrogen bonds per unit length of sequence, it could have Does this metabolic effect of energy flow have any effect in reference to the results obtained?

This is an interesting hypothesis, but we expect that selective pressures driving ARG dissemination would overcome the replicative cost effects. In addition (see below), the %GC content difference is not between the plasmid and the host; rather, it is mostly constrained to the ARG carried by the plasmid, so its increased replicative cost should be minimal.

How do you explain that the transfer of conjugative plasmids that requires specific recognition by the donor and recipient is independent of the %GC? Wouldn't it be more relevant to focus on the determinants that allow plasmid transfer than %GC?

The reviewer raises a very interesting point. The dissemination of ARGs to hosts with a very dissimilar %GC content could in principle be mediated by the proliferation of the plasmid harboring them in the new chromosomal background, raising issues about replication-associated metabolic costs and of conjugation efficiency. Alternatively, ARG dissemination could be mediated by transfer of the ARGs to a host-compatible plasmid, either individually (e.g. integron cassettes) or collectively (e.g. transposition). Analysis of plasmid vs. host chromosome %GC content for highly-disseminated genes shows strong correlation between plasmid and host %GC content, and therefore supports the latter scenario. We have included this in the discussion (Lines 452-471).

In lines 105-1028, it says

“In addition, our results show that ARG conferring resistance to specific antimicrobial classes present higher dissemination than genes conferring cross-resistance to multiple antibiotics. We report that different antimicrobial classes present remarkably diverse ARG dissemination profiles, and that plasmid-based ARG dissemination is not detected for many restricted-use antimicrobials.”

I understand that in this paragraph it would be worth clarifying that a specific dissemination event should be associated with the intensity of the selection pressure on the one hand and the resistance phenotype of the bacteria, since it would make evolutionary sense to promote a survival result. in the short term. On the other hand, from an evolutionary perspective it is logical to think that the adaptation mechanisms to multiple antimicrobials, or non-specific resistance mechanisms should be housed in the bacterial chromosome, which ensures greater stability over time and a vertical transfer assuming that it is information. genetics that have greater versatility and ecosystem functionality. I understand that this thesis, which is implicit in your reasoning, should be explicitly and adequately referenced. This thesis is later corroborated in the document, but is not referenced by “This indicates that efflux pumps conferring resistance to these compounds are present in most chromosomal backgrounds, from which they have been repeatedly disseminated.”

We agree that this is an important point to make and we have added text to the discussion to make this overarching assumption explicit (Lines 408-418). We have also referenced the statement concerning the presence of efflux pumps in most chromosomal backgrounds (Line 202).

You indicate that the Antibiotic resistance Ontology (ARO) was used in other works, arguing that it is a technique that restricts the analysis because it reduces the problem and makes it discrete, while your method incorporates a greater amount of data, however from my point of view Given the %GC can be associated with a huge amount of genetic information. I think that what would be interesting in this case would be to have focused his analysis on the GC content of the compatibility factors in the HGT phenomena associated with conjugative plasmids. I think it is a valid analysis, but very general. I believe that a paragraph should be included to clarify this point.

Our comment on AROs is that they are restrictive in that they subdivide known ARG homologs based on their original reporting in the literature. For instance, dfr genes are classified into multiple AROs, but they are all homologs. This is the motivation for grouping AROs mapping to the same COG (i.e. homologous group), as a way to analyze the behavior of homologous ARGs.

We agree with the reviewer that %GC analysis of plasmid incompatibility factors could provide interesting insights into the dissemination of these elements across plasmids, especially if using plasmid %GC as a control. This is beyond the scope of this work, which focuses on known ARGs, but we have included text to highlight that the techniques introduced here can be applied to analyze the dissemination behavior of other genes, such as incompatibility factors (Lines 546-549).

In the process of analyzing the sequences, I understand that there should be a difference between the sequences from bacteria that inhabit a wide diversity of ecological niches with respect to those bacteria circumscribed to much more specific ecosystem conditions (for example, sequences from extremophile bacteria, where the ecosystem conditions are much more stable), correlative analyzes between these two major categories of content would allow ratifying his thesis or explaining the generality of this phenomenon. In this sense, in their analysis they refer to Kleuverya, but it is a very widespread bacterium from the point of view of the variety of environmental factors to which it adapts. In baceria with less susceptibility to accept mobile elements foreign to the ecosystem, such as extremophile bacteria, would the specific elements associated with the transfer behave in the same way? Would they be determined by the %GC?

Our analysis focuses on genes and their dissemination profiles. Species with a broad habitat are generally exposed to a wider variety of environmental insults and interactions with other bacteria. They are thus more likely to harbor resistance genes and to act as potential sources for their dissemination. Extremophiles inhabit more secluded environments and are therefore less likely to be sources or recipients of disseminated ARGs, and the genetic transfer mechanisms involved might vary. Nonetheless, widespread dissemination of ARGs originating with or transferred to extremophiles would still be detectable via the %GC analysis introduced here.  

In another part of the document he says;

This indicates that the selective pressure underpinning the widespread dissemination of a particular ARG does not necessarily elicit its independent mobilization from different chromosomal backgrounds. This apparently counterintuitive result can be explained by multiple over- lapping rationales

In reference to this assertion it is convenient to consider the following, in the work it is assumed that the adaptive strategy of bacteria to face the presence of antibiotics is based on processes of independent dissemination of mobile elements that carry ARG, and for this it exemplifies a situation in where this phenomenon is not observed, but can be explained by overlapping. However, it is logical to think that bacterial genomic networks (gene regulatory network GRN or “chromosomal backgrounds”), by functioning in an intrinsically cooperative way in free-living bacteria and in those that are forming microcolonies (which is the ecosystem context where it is most favors HGT), modify their interactome after having integrated a mobile genetic element, resulting with greater or lesser efficiency in the adaptive process. In this sense, it would be worth considering that the phenomena of regulation of expression or regulation of transport (for example, porin-associated permeability, MFS or RND) is under the control of the bacterial chromosome, therefore there must be an interaction phenomenon between the pre-existing information in the bacterium and that of the conjugative plasmid that was integrated, generating an adaptive result that "cryptic" way (that is, through processes of regulation of expression or energy balances in metabolic pathways) generates a response in bacterial cell physiology that promotes an increase in the rate of mobilization. In this sense, it would be worth using this same methodology to determine the mobilization intensity of a siderophore for heavy metals, for example, and to determine its dispersion in a medium with a selection pressure imposed by the presence of antibiotics, for example. From my point of view, the modification of the interactome of the bacterium causes this process to be perceived as independent when it is not, but rather is induced by a cryptic mechanism.

We agree with the reviewer that the integration of the plasmid-encoded resistance genes with the host transport regulatory network can generate adaptive results through cryptic pathways that may promote mobilization. We have added text to reflect this (Lines 232-235). We also believe that our analysis platform could be used to study the dissemination behavior of other genes, such as siderophores, and we have noted this in the text (Lines 546-549).

In contrast, three resistance mechanisms (antibiotic inactivation, target replacement and target protection)  display remarkably high DI values (Figure 4)

In this statement, I believe it is convenient to emphasize that the mechanisms where a high rate of dissemination is perceived are specific processes, and therefore more energy efficient to adapt to the presence of antimicrobials. From an adaptive perspective, it is useful information for a reasonably short time, and therefore more susceptible to being used by bacteria through HGT phenomena that makes it possible, over generations, to eliminate said information from the genetic background, improving with they reproductive yields. In this sense, I think it is convenient to include this type of information with a bibliographical citation.

We agree with the reviewer that the specific mode of action of these mechanisms is likely related to their observed high dissemination and we have incorporated this in the discussion (Lines 281-285, 408-418).

Elsewhere in the document you can read;

ARGs conferring resistance via target alteration present an average DI of 22.0%, a significantly lower value than those discussed above

Referring to this result, it would be worth taking up the comment that I previously wrote in reference to the interactome, in this sense, the modification of a target requires a route from the receiving bacterium to integrate the information present in the plasmid, (for example a transcription factor, even the lack of a biochemical pathway), is not an autonomous or independent process within cell physiology, this determines that the elements that induce or promote dissemination require an element or specificity factor associated with the transfer in reference to the physiology of donor/recipient bacteria

We agree with the reviewer that resistance via target alteration may require adaptations in host biochemical pathways to integrate the target alteration, which can limit their ability to disseminate. We have added text to reflect this in the discussion (Lines 313-317).

Finally, it says in the document:

The apparently contradictory results for ARO-COG groups mapping to the target alteration mechanism are explained by the fact that this CARD Resistance Mechanism category includes both spontaneous mutations to targeted enzymes as well as enzymes that induce chemical modifications in the antimicrobial target protein

In this statement, I do not consider that the mutation rate is a factor that explains this result; Indeed, spontaneous mutations do occur, but what modifies the mutations are the protein-protein interaction network, which can be more or less efficient. This situation is different when the interaction does not occur, because the biochemical pathway is missing, or because the protein or enzyme that "chains" the resistance mechanism is not present. In other words, the capacity or potential for information capture through conjugative plasmids must necessarily be regulated by the receiving bacterium, in my opinion due to the potential combinatorial capacity of the plasmid information with the bacterial GRN.

Our statement does not imply an active role of different mutation rates in this process, but rather on the availability of spontaneous mutants that provide resistance and can be accessed via short mutational pathways. We agree that integration into the host biochemical pathways and physiological processes are also a limiting factor for target alteration mutations that can contribute to explain their reduced mobilization profiles, and we have incorporated text to reflect this (Lines 313-317).

Reviewer 3 Report

Sánchez-Osuna et. al, planned an in-silico study to address some important questions related to antimicrobial resistance and dissemination of related genes. More importantly, authors checked the efficacy of %GC content analyzer to monitor the dissemination of resistance determinants. However, there is no clear evidence for the set claims because of lack of statistical test and transparency. The key question, %GC content for the genes studied is missing. Conversely, the %GC content of the dissemination related genes has not been shown/ presented anywhere in the manuscript.

Lines 21-23: Line is long and need revision to be meaningful.

Line 87-88: In correct claims for the content of references, the cited studies 14, and 21 doesn’t conclude any such information about GC% normalization and million years of concept.

I suggest to use research papers to support such statements based on which this study is planned.

What is the proposed mechanism happens there for this alignment of %GC ? No such information is provided.

Lines 91-92: Figure 1A doesn’t show any such information written in these lines. This plot is simply saying the %ARG GC content is aligns with that of the host %GC content.

Line 322: The line states that "Antimicrobial class present higher dissemination index values"

but is that statistically significant ? I don’t see any statistical data in support of this claim. Even the claim is not universal in Figure 5 which demonstrates that for so many of the "Specific genes", DI is observed as low minimum.

Moreover, the table s4 is also missing the GC content of respective genes. Which should also be there if you say GC content as a determinant of ARG dissemination. What is correlation in GC and DI ?

Line 352: As written, Figure 6A doesn’t show such percentages.

Author Response

Sánchez-Osuna et. al, planned an in-silico study to address some important questions related to antimicrobial resistance and dissemination of related genes. More importantly, authors checked the efficacy of %GC content analyzer to monitor the dissemination of resistance determinants. However, there is no clear evidence for the set claims because of lack of statistical test and transparency. The key question, %GC content for the genes studied is missing. Conversely, the %GC content of the dissemination related genes has not been shown/ presented anywhere in the manuscript.

We have incorporated statistical testing as indicated by the reviewer and reworded the text to clarify the statement. The %GC content for all analyzed genes was provided in Supplementary Table S1. The text has been edited to make this explicit and facilitate locating the relevant data (Line 143).

Lines 21-23: Line is long and need revision to be meaningful.

The sentence has been reworded for clarity following suggestions from a native English speaker (Lines 22-24).

Line 87-88: In correct claims for the content of references, the cited studies 14, and 21 doesn’t conclude any such information about GC% normalization and million years of concept.

I suggest to use research papers to support such statements based on which this study is planned.

Lawrence and Ochman (1997) is the reference article regarding %GC amelioration in introgressed genes. Their work provides a mathematical framework for the evolution of %GC content following introgression, and analyzes numerous examples of gene %GC content amelioration in Escherichia coli, with estimated amelioration times of millions of years (see, for instance, Table 3 in the manuscript). In the second reference, we used Lawrence and Ochman’s model to estimate the required time for amelioration of folP genes transferred from the Leptospiraceae to the Rhodobiaceae, and we reported a lower bound of 476 million years for a 22 percentual unit change in %GC content. These are the relevant research papers providing support for the conceptual framework of this study.

What is the proposed mechanism happens there for this alignment of %GC ? No such information is provided.

Amelioration of %GC content in introgressed genes is driven by the mutational processes of the host. Over time, introgressed genes accumulate mutations following the host mutational biases, and may also undergo selection to match the host codon usage bias. Rates of divergence in %GC content for coding sequences have been estimated at 0.045% and 0.91% per million years at nonsynonymous and synonymous codons (Lawrence and Ochman, 1997). This explanation is now included in the text (Line 96-102).

Lines 91-92: Figure 1A doesn’t show any such information written in these lines. This plot is simply saying the %ARG GC content is aligns with that of the host %GC content.

Figure 1 illustrates the two possible theoretical scenarios for the relationship between ARG and chromosomal %GC content. As the reviewer notes, Figure 1A depicts the scenario in which ARGs have been mobilized (i.e. are detected in plasmids) in multiple lineages, but have not undergone widespread dissemination. Because the ARGs have been mobilized from phylogenetically close chromosomal backgrounds, their %GC content is expected to match the chromosomal %GC content, leading to alignment between ARG and chromosomal %GC content. The text has been amended to clarify this point (Lines 104-105).

Line 322: The line states that "Antimicrobial class present higher dissemination index values" but is that statistically significant ? I don’t see any statistical data in support of this claim. Even the claim is not universal in Figure 5 which demonstrates that for so many of the "Specific genes", DI is observed as low minimum.

We agree with the reviewer that the statement is not in agreement with the data shown in Figure 5. In line with Figure 6, we have amended Figure 5 to show the total number of homologs detected for each antimicrobial class (taking into account whether the CARD resistance gene associated with each homolog is designated as providing resistance to only one (specific) or more than one (multiple) antimicrobial classes), as well as the number of homologs in dissemination bands. Dissemination index percentages are superimposed. The plot shows that disseminated genes providing resistance to specific antimicrobial classes significantly outnumber those providing resistance to multiple antimicrobial classes. We have reworded the sentence (Line 349) and included the result of a one-tailed Mann-Whitney test to substantiate the claim (Line 351).

Moreover, the table s4 is also missing the GC content of respective genes. Which should also be there if you say GC content as a determinant of ARG dissemination. What is correlation in GC and DI ?

Table S4 shows the the dissemination index, number of detected homologs and number of homologs in dissemination bands for different antimicrobial classes and resistance mechanisms, taking into account whether the CARD resistance gene associated with each homolog is designated as providing resistance to only one (specific) or more than one (multiple) antimicrobial classes. The %GC content for all homologs is reported in Supplementary Table S1.

We do not state that %GC content is a determinant of ARG dissemination. We simply note that the %GC content of a broadly disseminated gene can be used to assess its dissemination because it is effectively independent of the %GC content of the hosts among which it has been disseminated. There is no meaningful correlation between %GC and DI (Pearson r = -0.16). The %GC of the ARG is determined by its chromosomal origin, and the DI by the presence of homologs of the ARG (with nearly identical %GC) in plasmids harbored by species with different chromosomal %GC content.

Line 352: As written, Figure 6A doesn’t show such percentages.

The reviewer is correct. We have added the dissemination index percentages to Figure 6.

Round 2

Reviewer 2 Report

Once the document has been reviewed, I consider that all my suggestions were attended to, included and adequately referenced. It is quality and interesting work.

Author Response

Once the document has been reviewed, I consider that all my suggestions were attended to, included and adequately referenced. It is quality and interesting work.

We thank the reviewer for their insightful comments and suggestions, which we believe have greatly improved the manuscript.

Reviewer 3 Report

Methods / strategies of work are not clearly stated in manuscript which is cause of several questions about the methodology used to make figures, tables and conclusions. I still have multiple concerns on this study (specifically about %GC, Figure 1, Dissemination Index and new conclusions made about ARG homologs outnumbering in specific genes). I hope resolving these concerns may improve the manuscript.

Comment No. 1:

What type of study it is must be reflected in the title itself:

I recommend modifying title of this study as: “In-silico systematic assessment of antimicrobial resistance dissemination across 2 the global plasmidome”. Abstracts are unstructured for this journal but there must be a sequence of content in the abstract: short background/ objective of study, methods used (please clearly state that this is an in-silico study), results / key findings and then conclusions. There are several modifications done in manuscript regarding the statements/ claims about %GC, DI, and regarding DI of specific genes. Please remove all those statements from abstract, conclusion and other parts of manuscript, or change these statements by saying that further wet-lab works needed to validate these in-silico findings. Now, you have revised your statements and shifted the said claims from DI to the outnumbered homologs: WHICH IS ALSO NOT TRUE. I just cross-checked your data for antibiotic class “sulfonamide antibiotic” in Table S4. Which says genes marked as “MULTIPLE” have number of ARG homologs as “3086, 839, 35 and 2212”, which is much more lower than the “SPECIFIC” ones (these numbers are 75, 75 in case of specific categories). Even these new claims are not authentic and consistent.

Comment 2:

Figure 1: Please add relevant data in tabular for, what are the names of genes which you have classified as Non-Disseminated ARG and those defined as Disseminated ARGs too. % GC for each one must also be mentioned for each gene in a separate column.

Also clarify, which parameter you have used to classify those genes into two categories (Non-disseminated ARG and Disseminated ARG) which are (Figure 1). The information provided in Table S5 is not appropriate, as the plasmid accession ARG homolog ID doesn’t go anywhere when searching google or NCBI. Have you made this classification based on the DI index ? If yes, it is not realistic or experimentally proven. How can you classify dissemination potential of genes based on only an computational index without having any correlation with experimental data. So down the lane, claims made may not be authentic.

Comment 3:

Your reply to one of the comment is: We do not state that %GC content is a determinant of ARG dissemination. We simply note that the %GC content of a broadly disseminated gene can be used to assess its dissemination because it is effectively independent of the %GC content of the hosts among which it has been disseminated. There is no meaningful correlation between %GC and DI (Pearson r = -0.16). The %GC of the ARG is determined by its chromosomal origin, and the DI by the presence of homologs of the ARG (with nearly identical %GC) in plasmids harbored by species with different chromosomal %GC content. Moreover, Figure 1 is the basis of %GC based claims as dissemination marker, which is already under question in comment 2. Please address for clarifications. May the methods could be written with clarity, there could not have been this many questions on manuscript.

Comment 4:

The statement regarding %GC as powerful method to monitor dissemination of ARGs are still there in manuscript and in abstract.

However, you have stated in your reply that there is no correlation between %GC content and DI. In addition, you reply also mentioned that “you simply note that %GC content can be used to assess it’s dissemination”. This is only a general observation which is true at some places and also false at instances in your data. A general observation without a well-planned controlled wet-lab experiment, needs to be carefully stated anywhere in manuscript and may not be mentioned in abstract or conclusions. If these statements are important to be mentioned they can be stated by saying that experimental/ wet lab investigations are needed to validate these preliminary results. Else, I suggest to remove all such statements from the manuscript since they are not authentic.

Comment 5:

Authors have used / defined a new parameter to determine the dissemination of genes and it is Dissemination Index (DI). DI don’t even have correlation with %GC in your data. Or does your tool predicts high dissemination potential in the genes which are proved to really highly disseminated in literature. Can you please validate your DI tool on the existing highly disseminated genes reported in literature along-with comparing with the genes having low dissemination rates. I understand that you have used mathematical formulae to calculate dissemination index (DI) of each ARG in a specific class but how much your DI calculations are realistic ? Needs experimental validations atleast for 6 most disseminated genes reported in literature compared to non-disseminated ones.

Comment 6:

Conclusions must be carefully written in context to %GC and DI. The paragraph may be restructured.

Author Response

Methods / strategies of work are not clearly stated in manuscript which is cause of several questions about the methodology used to make figures, tables and conclusions. I still have multiple concerns on this study (specifically about %GC, Figure 1, Dissemination Index and new conclusions made about ARG homologs outnumbering in specific genes). I hope resolving these concerns may improve the manuscript.

We thank the reviewer for pointing out aspects of the manuscript that require clarification. We have addressed the reviewer’s concerns in the text, as detailed in the responses below.

Comment No. 1:

What type of study it is must be reflected in the title itself:

I recommend modifying title of this study as: “In-silico systematic assessment of antimicrobial resistance dissemination across 2 the global plasmidome”. Abstracts are unstructured for this journal but there must be a sequence of content in the abstract: short background/ objective of study, methods used (please clearly state that this is an in-silico study), results / key findings and then conclusions. There are several modifications done in manuscript regarding the statements/ claims about %GC, DI, and regarding DI of specific genes. Please remove all those statements from abstract, conclusion and other parts of manuscript, or change these statements by saying that further wet-lab works needed to validate these in-silico findings. Now, you have revised your statements and shifted the said claims from DI to the outnumbered homologs: WHICH IS ALSO NOT TRUE. I just cross-checked your data for antibiotic class “sulfonamide antibiotic” in Table S4. Which says genes marked as “MULTIPLE” have number of ARG homologs as “3086, 839, 35 and 2212”, which is much more lower than the “SPECIFIC” ones (these numbers are 75, 75 in case of specific categories). Even these new claims are not authentic and consistent.

We have modified the title according to the reviewer’s suggestion. The total number of detected disseminated genes conferring resistance to specific antimicrobial classes is larger than the one for genes conferring resistance to multiple antimicrobial classes (49.3% vs. 35.9%). We agree with the reviewer, however, that there are clear exceptions to this trend (such as the sulfonamides/sulfones, which CARD classifies as distinct drug classes as explained in the text), and that therefore the statement is equivocal. We have therefore removed the statement from the results and other sections of the text (Lines 22-24, 349-352, 408-410, 561-562).

Comment 2:

Figure 1: Please add relevant data in tabular for, what are the names of genes which you have classified as Non-Disseminated ARG and those defined as Disseminated ARGs too. % GC for each one must also be mentioned for each gene in a separate column.

This is an illustrative example figure and, therefore, not based on real data. We have modified the text and figure legend to make this point clear.

Also clarify, which parameter you have used to classify those genes into two categories (Non-disseminated ARG and Disseminated ARG) which are (Figure 1). The information provided in Table S5 is not appropriate, as the plasmid accession ARG homolog ID doesn’t go anywhere when searching google or NCBI. Have you made this classification based on the DI index ? If yes, it is not realistic or experimentally proven. How can you classify dissemination potential of genes based on only an computational index without having any correlation with experimental data. So down the lane, claims made may not be authentic.

Again, Figure 1 is an illustrative example and therefore not based on real data. We have tested the accession numbers listed in Table S5 (now Table S6) and they are all valid. For instance, in row 1 of Table S5 we find:

WP_001287391 - accessible at https://www.ncbi.nlm.nih.gov/protein/WP_001287391

NZ_MN540571 - accessible at https://www.ncbi.nlm.nih.gov/nuccore/NZ_MN540571 

Comment 3:

Your reply to one of the comment is: We do not state that %GC content is a determinant of ARG dissemination. We simply note that the %GC content of a broadly disseminated gene can be used to assess its dissemination because it is effectively independent of the %GC content of the hosts among which it has been disseminated. There is no meaningful correlation between %GC and DI (Pearson r = -0.16). The %GC of the ARG is determined by its chromosomal origin, and the DI by the presence of homologs of the ARG (with nearly identical %GC) in plasmids harbored by species with different chromosomal %GC content. Moreover, Figure 1 is the basis of %GC based claims as dissemination marker, which is already under question in comment 2. Please address for clarifications. May the methods could be written with clarity, there could not have been this many questions on manuscript.

We state again that %GC content is not a determinant of ARG dissemination. The %GC content of an ARG is determined by the %GC content of its chromosomal origin. As explained in the manuscript following this reviewer’s suggestion %GC content takes millions of years to ameliorate. Therefore, when an ARG is mobilized to a plasmid and disseminates broadly among bacteria with different chromosomal %GC content it is possible to detect this dissemination by analyzing the correlation between the ARG (essentially invariant) %GC content and the %GC content of the host on which the plasmid harboring the ARG has been isolated. We have edited the text accompanying Figure 1 as well as the methods to articulate this better.

Comment 4:

The statement regarding %GC as powerful method to monitor dissemination of ARGs are still there in manuscript and in abstract.

However, you have stated in your reply that there is no correlation between %GC content and DI. In addition, you reply also mentioned that “you simply note that %GC content can be used to assess it’s dissemination”. This is only a general observation which is true at some places and also false at instances in your data. A general observation without a well-planned controlled wet-lab experiment, needs to be carefully stated anywhere in manuscript and may not be mentioned in abstract or conclusions. If these statements are important to be mentioned they can be stated by saying that experimental/ wet lab investigations are needed to validate these preliminary results. Else, I suggest to remove all such statements from the manuscript since they are not authentic.

This is not a general observation. The analysis is based on well-established evolutionary dynamics for the %GC content of introgressed genes as well as previous published work showing that the relationship between the %GC content of plasmid-borne ARG homologs and that of their isolation hosts chromosomal %GC can identify dissemination events. Here we generalize the analysis to all ARGs available in CARD and all plasmid sequences available in PLSDB. We have noted the need for follow-up experimental investigation in the conclusions.

Comment 5:

Authors have used / defined a new parameter to determine the dissemination of genes and it is Dissemination Index (DI). DI don’t even have correlation with %GC in your data. Or does your tool predicts high dissemination potential in the genes which are proved to really highly disseminated in literature. Can you please validate your DI tool on the existing highly disseminated genes reported in literature along-with comparing with the genes having low dissemination rates. I understand that you have used mathematical formulae to calculate dissemination index (DI) of each ARG in a specific class but how much your DI calculations are realistic ? Needs experimental validations atleast for 6 most disseminated genes reported in literature compared to non-disseminated ones.

There is no rationale for DI having to correlate with %GC. The %GC is determined by the chromosomal source of an ARG. The DI is determined by the relationship between the %GC of plasmid-borne ARG homologs and that of the plasmid isolation hosts.

Our analysis identifies 29 instances of broad ARG dissemination that have been previously reported in the literature. Many of them are discussed in the text, but this result is now stated in the text and summarized in Table S3 (Lines 162-165). Beyond its theoretical foundation, this provides ample validation for the ability of this method to identify and quantify ARG dissemination.

Comment 6:

Conclusions must be carefully written in context to %GC and DI. The paragraph may be restructured.

We have reworded the conclusions according to the reviewer’s comments.